# Unsupervised Representation Transfer for Small Networks: I Believe I Can Distill On-the-Fly

Hee Min Choi, Hyoa Kang, and Dokwan Oh

Samsung Advanced Institute of Technology
Suwon, South Korea

{chm.choi, hyoa.kang, dokwan.oh}@samsung.com

## Abstract

A current remarkable improvement of unsupervised visual representation learning is based on heavy networks with large-batch training. While recent methods have greatly reduced the gap between supervised and unsupervised performance of deep models such as ResNet-50, this development has been relatively limited for small models. In this work, we propose a novel unsupervised learning framework for small networks that combines deep self-supervised representation learning and knowledge distillation within one-phase training. In particular, a teacher model is trained to produce consistent cluster assignments between different views of the same image. Simultaneously, a student model is encouraged to mimic the prediction of on-the-fly self-supervised teacher. For effective knowledge transfer, we adopt the idea of domain classifier so that student training is guided by discriminative features invariant to the representational space shift between teacher and student. We also introduce a network driven multi-view generation paradigm to capture rich feature information contained in the network itself. Extensive experiments show that our student models surpass state-of-the-art offline distilled networks even from stronger self-supervised teachers as well as top-performing self-supervised models. Notably, our ResNet-18, trained with ResNet-50 teacher, achieves 68.3% ImageNet Top-1 accuracy on *frozen* feature linear evaluation, which is only 1.5% below the supervised baseline.

## 1 Introduction

Recently, there has been a growing attention in unsupervised and self-supervised learning where the goal is to effectively learn useful features from a large amount of unlabeled data. Current self-supervised visual representation learning methods appear to approach and possibly even outperform the fully-supervised counterpart [8, 25, 29]. All top-performing self-supervised learning (SSL) algorithms for visual representation involve training deep models on powerful computers. In particular, their smallest architecture is ResNet-50 [26], and the networks are trained with large batches (e.g., 4096 images) on multiple specialized hardware devices such as 128 TPU cores [6, 8, 25]. Yet, this heavy implementation is not a viable option in a resource-limited environment, and there is evidence that the SSL methods even do not work well on light models (Figure 1 and [22]). Also, we need strong small networks that can operate on a system on a chip (SoC) for real-world applications.

Existing methods on SSL have demonstrated that deeper models learn general visual representation more effectively with unlabeled data [8, 40]. Moreover, it has been empirically shown that predictive performance of bigger networks is better transferred into smaller ones [9]. Inspired by these, we are

35th Conference on Neural Information Processing Systems (NeurIPS 2021).

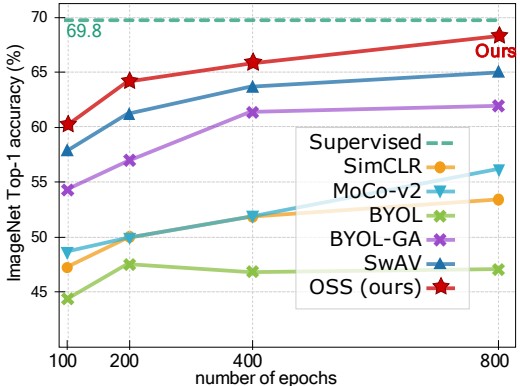

Figure 1: ImageNet Top-1 accuracy of linear probe on ResNet-18 representations obtained from various self-supervised methods with 256-batch training. BYOL-GA denotes the implementation of BYOL with gradient accumulation (to 4096 batches). Green dotted line indicates supervised ResNet-18. Our method (red star) surpasses state-of-the-art self-supervised methods.

interested in pretraining a light-weight model by distilling representational knowledge from a deep self-supervised network instead of directly performing self-supervised training on a small one.

Most of previous distillation methods in unsupervised representation learning literature are offline in the sense that they leverage an already trained self-supervised teacher to transfer feature information to students [22, 31, 40, 57]. Often, their sequential training pipelines require extra pre/post-processing steps in order to boost performance. Unlike these approaches, we propose a novel unsupervised representation learning framework for small networks that combines deep self-supervised training and knowledge distillation [30] within one-phase training. To be specific, our teacher learns clusters and representation, and at the same time the student is trained to align its prediction to the clustering of **O**n-the-fly **S**elf-**S**upervised teacher. From this, we refer to our method as OSS. The main advantage over offline distillation is that our teacher better distills proper signals into the student at each training stage by solving the same task online. In distillation process, it is difficult for a low-capacity student to perfectly mimic the large teacher's behavior [36]. To deal with this, we incorporate the idea of domain classifier [23]. That is, we add a *feature classifier* trained not to be able to distinguish between teacher and student embeddings, and this leads to the emergence of representational-space-invariant features in the course of joint training.

There is evidence that increasing the number of different views improves the quality of resulting features in SSL [6, 8, 55]. Existing approaches to multi-view generation rely on random image transformation techniques. Different from these methods, we introduce a new network driven paradigm in order to utilize rich feature information contained in the network itself. In particular, we apply random dropout and empirically show that this apparently improves label efficiency (Section 5).

Our method is conceptually simple but surprisingly works well with typical 256-batch training on a single 8-GPU machine. Extensive experiments show that our student networks outperform state-of-the-art self-supervised models (Figure. 1) as well as students from top-performing distillation approaches even with stronger self-supervised teachers (Tables 1 and 5) on linear evaluation. We also evaluate learned representations on series of vision tasks with multiple network architectures and various benchmark datasets and demonstrate significant performance gain (Tables 2, 3 and 4 and Figure 3). Overall, we make the following contributions.

- We propose a novel unsupervised visual representation learning framework for small networks that simultaneously conducts self-supervised teacher training and knowledge distillation.

- We introduce a network driven view generation paradigm and an adversarial distillation scheme to better capture representational information from networks. This is the first work to show their effectiveness in the context of unsupervised visual representation learning.

- The proposed method significantly improves unsupervised pretraining performance of small models, and its effectiveness is shown under multiple settings of network architectures, datasets and vision tasks.

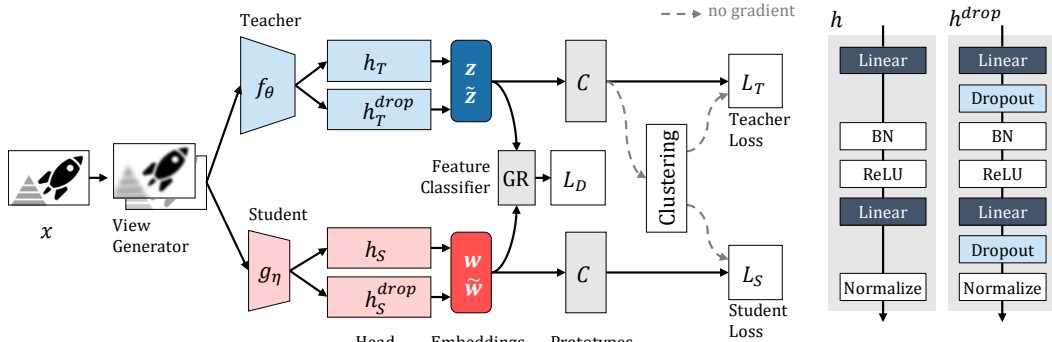

Figure 2: **Overview of proposed architecture.** OSS framework takes as input two randomly augmented views $x_1$ and $x_2$, and they are processed by teacher's and student's backbones $f_\theta$ and $g_\eta$. The processed outputs are then projected to lower dimensional embeddings $z$ and $w$ via teacher's and student's projection heads $h_T$ and $h_S$. We add dropout to each linear layer in both projection heads ($h^{\mathrm{drop}}$'s) for generating additional views of embeddings $\tilde{z}$ and $\tilde{w}$. We note that $h$ and $h^{\mathrm{drop}}$ share weights. Network training is guided by cluster prediction mechanism, and the target cluster assignments are obtained by online clustering using the mapping of teacher's feature embeddings $z$ to prototype vectors $C$. To distill useful discriminative feature information invariant to the representational space shift between teacher and student, we use a feature classifier connected to the last layer of each projection head with a gradient reversal layer (GR).

## 2 Related Work

**Unsupervised visual representation learning.** In unsupervised or self-supervised visual representation learning, we learn rich feature information by solving pretext tasks whose labels can be obtained from image data itself. Learning methods can be divided into several categories by types of pretext tasks used therein: construction-based methods [43, 62], prediction-based methods [10, 20, 24, 25, 38, 39], cluster-based methods [3, 5, 6], and contrastive methods [8, 11, 29]. Our work is mostly related to [6], a recent clustering-based method.

**Multi-view generation.** Contrastive and clustering-based self-supervised visual representation learning methods exploit random image transformation techniques to generate different views of each image. This is because comparing more views during training improves the performance of the resulting model [6, 8, 37, 55]. Our method uses dropout [50] for obtaining more views to better utilize useful feature information as well as applying existing data augmentation based strategies.

**Knowledge distillation from self-supervised models.** Most of knowledge distillation [30] techniques are proposed in the context of supervised learning, where a shallow network is trained via supervision signals from both labels and a deep network [1, 41, 44, 47, 52, 53, 58]. In unsupervised learning settings, recent studies [40] and [57] exploit clustering of teacher's embeddings as distillation signals. More recently, CompRess [31] maintains queues for storing data sample's embeddings and calculates instance-level similarity scores between all the samples in the queue and the input features extracted from teacher (and student). Its training objective is to minimize the distance between teacher and student score distributions. SEED [22] is a similar concurrent work of [31]. Unlike aforementioned methods that rely on already trained static teachers, our framework injects a distillation step into clustering-based self-supervised teacher learning. The only existing online technique is [4], which combines contrastive learning and co-teaching.

**Domain adversarial learning.** DANN [23] learns features that are discriminative for the main classification task and indiscriminate with respect to the domain shift. This is achieved by adding a domain classifier with a gradient reversal layer. In particular, network parameters are updated to minimize the main classifier loss and to maximize the domain classifier loss via a gradient reversal layer that reverses the sign of gradient. We adopt this idea in our framework to transfer discriminative features invariant to the representational space change between teacher and student. Similar idea with a generative adversarial network [35] is applied in supervised knowledge distillation in [12].

# 3   Proposed Framework

Our goal is to learn visual features of small networks effectively without access to any labeled data. In order to do this, we embed a distillation step into the process of on-the-fly self-supervised teacher learning. Inspired by [57], which empirically shows clustering improves generalization power of representation, our online framework is developed upon [6], the state-of-the-art clustering-based self-supervised method. We also incorporate the idea of domain classifier to facilitate learning of useful features invariant to the representational space change between teacher and student. In a nutshell, the training objective function of our network is given by

$$L = L_T + L_S + L_D, \tag{1}$$

where $L_T$, $L_S$, and $L_D$ are designed for teacher network training (Sections 3.1 and 3.2), knowledge distillation for student network (Sections 3.1 and 3.2), and domain classifier (Section 3.3), respectively. Overall architecture is presented in Figure 2, and a pseudo code is available in A.1.

## 3.1   Teacher training and knowledge distillation

We now outline teacher network (e.g. ResNet-50) training and knowledge distillation framework. Our architecture takes as input two randomly augmented views $x_{n1}$ and $x_{n2}$ of an image $x_n$. The two views are processed by teacher's backbone network $f_\theta$ and a projection head $h_T$ that consists in a multi-layer perceptron (MLP). The processed features are then projected to the unit sphere. We denote teacher's backbone features as $f_{n1} = f_\theta(x_{n1})$ and $f_{n2} = f_\theta(x_{n2})$, and the projected outputs as $z_{n1} = \frac{h_T(f_{n1})}{\|h_T(f_{n1})\|_2}$ and $z_{n2} = \frac{h_T(f_{n2})}{\|h_T(f_{n2})\|_2}$. Also, the two images are feed-forwarded to student's backbone network $g_\eta$ (e.g., ResNet-18). The outputs are projected via an MLP head $h_S$ and then normalized as well. Similarly, we denote student's backbone features as $g_{n1} = g_\eta(x_{n1})$ and $g_{n2} = g_\eta(x_{n2})$, and their projected output vectors as $w_{n1} = \frac{h_S(g_{n1})}{\|h_S(g_{n1})\|_2}$ and $w_{n2} = \frac{h_S(g_{n2})}{\|h_S(g_{n2})\|_2}$.

We randomly initialize prototype vectors $C = \{c_1, \ldots, c_K\}$, where $K$ is the number of clusters. Code vectors, $q_n$'s, for cluster assignments are computed from mapping teacher's embedding representation $z_n$'s to the trainable prototype vectors $C$ and then solving an optimization problem for online clustering. We then solve a swapped prediction problem [6] for teacher and simultaneously distill teacher's knowledge into student by enforcing a consensus in prediction.

**Teacher training.** A training objective for the teacher network is to solve a swapped cluster prediction problem [6] of predicting the code $q_{n1}$ from the feature $z_{n2}$ and $q_{n2}$ from $z_{n1}$:

$$L_t(z_1, z_2) = \sum_n \left[ L_c(z_{n1}, q_{n2}) + L_c(z_{n2}, q_{n1}) \right]. \tag{2}$$

Here, $L_c$ is the cross entropy loss between the codes and softmax probabilities (with a temperature parameter $\tau$) of dot products of $z_{ni}$ and all prototypes in $C$:

$$L_c(z_{n1}, q_{n2}) = - \sum_k q_{n2}^{(k)} \log p_{n1}^{(k)} \quad \text{where} \quad p_{n1}^{(k)} = \frac{\exp\left(\frac{1}{\tau} z_{n1}^\top c_k\right)}{\sum_{k'} \exp\left(\frac{1}{\tau} z_{n1}^\top c_{k'}\right)}. \tag{3}$$

The loss function (2) is jointly minimized with respect to the prototypes $C$ and the parameters of teacher backbone $f_\theta$ and projection head $h_T$ implicitly contained in $z_n$'s. The codes $q_n$'s are computed using Sinkhorn-Knopp algorithm [18] under an equi-partition constraint for each augmented mini-batch with stored embeddings as done in [6].

**Knowledge distillation.**   The student model is trained to follow teacher's code assignment. The training loss has exactly the same form of (2) with features from student and targets from teacher's codes:

$$L_s(w_1, w_2) = \sum_n \left[ L_c(w_{n1}, q_{n2}) + L_c(w_{n2}, q_{n1}) \right]. \tag{4}$$

Specifically, the cross-entropy loss $L_c$ is similarly calculated using student's embedding vectors $w_n$'s and the prototype vectors $C$. Again, minimization of this loss is performed with respect to the prototypes $C$ and the parameters of student backbone $g_\eta$ and projection head $h_S$ used to produce $w_n$'s.

## 3.2 Network driven multi-view generation

In addition to applying multi-crop strategy [6], we propose a network driven multi-view generation method to utilize rich feature information contained in the network. In particular, we add a dropout layer after each linear layer in the projection heads to generate more feature embeddings. Additional loss terms from small-crop features $\{\hat{z}_{nv}\}_{v=1}^{V}$ and dropout features $\{\tilde{z}_{nv}\}_{v=1}^{V+2}$ are given by

$$L_{\mathrm{mv}}(\tilde{z}, \hat{z}) = \sum_{n} \sum_{i \in \{1,2\}} \sum_{v} \left[ L_c(\tilde{z}_{nv}, q_{ni}) + L_c(\hat{z}_{nv}, q_{ni}) \right], \tag{5}$$

where we note that codes $q_n$'s are computed from non-dropout full-crop features. To summarize, the teacher loss $L_T$ in the training objective (1) is the sum of (2) and (5):

$$L_T(z_1, z_2, \tilde{z}, \hat{z}) = L_t(z_1, z_2) + L_{\mathrm{mv}}(\tilde{z}, \hat{z}). \tag{6}$$

The corresponding loss $L_S$ for student in (1) is similarly computed by the sum of (4) and $L_{\mathrm{mv}}(\tilde{w}, \hat{w})$ obtained from student's small-crop and dropout generated features $(\tilde{w}, \hat{w})$ in place of teacher's $(\tilde{z}, \hat{z})$ using (5).

## 3.3 Feature classifier

In order to transfer good features that are discriminative for the main clustering task and indiscriminate with respect to the representational space shift between teacher and student, we adopt the idea of domain classifier [23], and we call it *feature classifier*. The feature classifier is adversarially trained not to be able to identify from which the embedding is originated via a gradient reversal layer. To be specific, the feature classifier $D$ consists in an MLP that projects the features to a 2-D space. We compute the cross entropy loss between the feature label (0 for teacher and 1 for student) and softmax probability of the feature classifier outputs given by

$$L_D(z, w) = -\sum_{n} \sum_{i} \left( \log d_{T,ni}^{(0)} + \log d_{S,ni}^{(1)} \right) \quad \text{where} \quad d_{T,ni}^{(k)} = \frac{\exp\left(D^{(k)}(z_{ni})\right)}{\sum_{k' \in \{0,1\}} \exp\left(D^{(k')}(z_{ni})\right)} \tag{7}$$

and $d_{S,ni}^{(k)}$ is similarly obtained as $d_{T,ni}^{(k)}$ with $w_{ni}$ in place of $z_{ni}$. For both teacher and student, the feature classifier is connected to the last layer of the MLP projection head ($h_T$ and $h_S$) via a gradient reversal layer that scales the gradient by a certain negative constant $\alpha$ during backpropagation training. Thus, minimizing $L_D$ reduces discriminative power of the feature classifier.

# 4 Experiments

We perform extensive experiments across different combinations of teacher-student architecture pairs and various tasks and datasets to test the performance of our student network. In this section, we focus on evaluating our ResNet-18 features trained with ResNet-50 teacher and a batch size of 256 for 800 epochs on ImageNet-1K [19] training set (see A.1 for details). For self-supervised ResNet-18

Table 1: Top-1 accuracy (%) of linear probe on ImageNet for ResNet-18 distilled from SwAV's ResNet-50. * indicates the final accuracy of online teacher.

| Distillation Method | Teacher Top-1 | Student Top-1 |
|---|---|---|
| Supervised | - | 69.8 |
| CC [40] | 75.6 | 60.8 |
| CRD [52] | 75.6 | 58.2 |
| SEED [22] | 75.3 | 62.6 |
| CompRess-2q [31] | 75.6 | 62.4 |
| CompRess-1q [31] | 75.6 | 65.6 |
| OSS (ours) | 73.0* | **68.3** |

Table 2: Top-1 accuracy (%) of ResNet-18 fine-tuned on small subsets of ImageNet. Our method outperforms both supervised and all considered unsupervised methods.

| Method | Top-1 Accuracy | |
|---|---|---|
| | 1% | 10% |
| Scratch | 14.9 | 50.4 |
| SimCLR [8] | 28.8 | 54.2 |
| MoCo-v2 [11] | 25.2 | 54.1 |
| BYOL [25] | 25.6 | 52.3 |
| SwAV [6] | 39.7 | 60.4 |
| CompRess-1q [31] | 37.8 | 59.0 |
| OSS (ours) | **47.8** | **63.6** |

Table 3: Transfer performance (%) of ResNet-18 finetuned on small-scale datasets for varying tasks. The evaluation measure is $AP_{50}$ (average precision with intersection over union threshold of 50%) for detection and instance segmentation and mIoU (mean intersection over union metric) for semantic segmentation. Overall, our model works best across all transfer tasks and datasets.

| Method | $AP_{50}^{bb}$ | | | $AP_{50}^{inst}$ | | mIoU |
| | VOC07 | COCO-1K | COCO-10K | COCO-1K | COCO-10K | VOC12 |
|---|---|---|---|---|---|---|
| Supervised | 66.9 | 17.0 | 33.6 | 15.6 | 31.1 | 42.2 |
| SimCLR [8] | 66.1 (-0.8) | 16.0 (-1.0) | 33.0 (-0.6) | 14.8 (-0.8) | 30.6 (-0.5) | 40.6 (-1.6) |
| MoCo-v2 [11] | 65.4 (-1.5) | 11.0 (-6.0) | 28.8 (-4.8) | 9.9 (-5.7) | 26.9 (-4.2) | 41.3 (-0.9) |
| BYOL [25] | 64.9 (-2.0) | 14.3 (-2.7) | 30.4 (-3.2) | 12.9 (-2.7) | 28.0 (-3.1) | 39.9 (-2.3) |
| SwAV [6] | 68.1 (+1.2) | 18.6 (+1.6) | 36.8 (+3.2) | 17.2 (+1.6) | 34.4 (+3.3) | 42.1 (-0.1) |
| CompRess-1q [31] | 66.4 (-0.5) | 14.8 (-2.2) | 31.7 (-1.9) | 13.5 (-2.1) | 29.4 (-1.7) | 42.0 (-0.2) |
| OSS (ours) | **68.3** (+1.4) | **20.0** (+3.0) | **37.9** (+4.3) | **18.4** (+2.8) | **35.4** (+4.3) | **43.2** (+1.0) |

baselines, we use top-performing SSL models of SimCLR [8], MoCo-v2 [11], BYOL [25], and SwAV [6] that are trained with 256 batches for 800 epochs in `pytorch` [42] unless explicitly specified (see A.2). By default, the supervised performance comes from official `torchvision` models.

## 4.1 Linear evaluation on ImageNet

We validate our method by linear evaluation of *frozen* features, following the protocol [6] described in A.3. We report 1-crop, top-1 classification accuracy of ResNet-18 on ImageNet validation set. Figure 1 shows that OSS significantly surpasses all the state-of-the-art SSL methods. Note that OSS requires shorter training epochs than the SSL algorithms to promise better performance. Indeed, our 200-epoch model (64.1%) is almost on par with 800-epoch SwAV's (64.9%). Moreover, our 800-epoch model achieves 68.3%, which is only 1.5% below the accuracy of the supervised model. A clustering-based SwAV [6] works better than contrastive methods of MoCo-v2 [11] and SimCLR [8]. We conjecture this is because clustering is relatively easier than instance discrimination for light networks in a large-scale dataset. BYOL-GA indicates the implementation of BYOL [25] with gradient accumulation (GA) to 4096 batches (see Appendix G2 of [25]). For a fair comparison, we will use its non-GA version (BYOL) throughout the following experiments as all other methods are trained using 256 batches without GA.

In Table 1, we also compare our approach with various offline distillation methods using SwAV's ResNet-50 teachers obtained from [31] and [22]. We note that [31] uses two extra normalization layers in the classifier for linear evaluation. Nevertheless, all the offline compression models from stronger SwAV's ResNet-50 teacher do not still match our student's performance. A plausible reason for this is that large models easily overfit to the training data, so the static teachers provide less extra knowledge beyond hard annotations [2]. A summary of the offline techniques along with an extensive list of their performance is found in B.1. We also test general representational power of the pretrained frozen encoders on other classification datasets in B.2.

## 4.2 Data efficiency evaluation on ImageNet

We evaluate the performance on ImageNet classification task when finetuning our 800-epoch pre-trained ResNet-18 model with small subsets of trainset using labels. We follow semi-supervised protocol [6] detailed in A.4, and use 1% and 10% splits of [8]. We report top-1 accuracy on ImageNet validation set in Table 2. Our method outperforms self-supervised approaches for all label fractions. We note that the gap in 1% data regime is larger than the gap in 10% counterpart. We also carry out the same data efficiency evaluation of ResNet-18 features obtained from the state-of-the-art offline knowledge distillation method of CompRess-1q [31] with SwAV [6]'s ResNet-50 teacher. We use the officially released model and find that our student significantly outperforms it by +10.0% in 1% and +4.6% in 10% settings. In fact, our network driven multi-view generation significantly improves label efficiency, and we will investigate this in ablations. Also, another strong low-shot learning performance on Places365 [63] dataset is available in B.3.

Table 4: Top-1 accuracy (%) of linear evaluation on ImageNet for various shallow models. The second column (Params) is the number of parameters in each backbone network, and the unit M denotes $10^6$. Params and supervised performance are obtained from the corresponding papers.

| Model | Params | Method | Top-1 |
|---|---|---|---|
| ResNet-18 [26] | 11.5M | Supervised | 69.8 |
| | | SwAV [6] | 61.2 |
| | | OSS (ours) | 64.1 |
| RegNetY-600MF [45] | 6.1M | Supervised | 75.5 |
| | | SwAV [6] | 67.0 |
| | | OSS (ours) | 67.4 |
| EffcientNet-B0 [51] | 5.3M | Supervised | 77.1 |
| | | SwAV [6] | 59.3 |
| | | OSS (ours) | 64.1 |
| MobileNet-v2 [48] | 3.4M | Supervised | 72.0 |
| | | SwAV [6] | 63.2 |
| | | OSS (ours) | 66.1 |

Table 5: Top-1 accuracy (%) of linear probe on ImageNet for MobileNet-v2 distilled from MoCo-v2's ResNet-50 teacher. * indicates the final accuracy of online teacher. Our method outperforms all considered offline approaches.

| Distillation Method | Teacher Top-1 | Student Top-1 |
|---|---|---|
| Supervised | - | 72.0 |
| CC [40] | 70.8 | 59.2 |
| CRD [52] | 70.8 | 54.1 |
| Reg [60] | 70.8 | 48.0 |
| Reg-BN [31] | 70.8 | 62.3 |
| CompRess-1q [31] | 70.8 | 63.0 |
| CompRess-2q [31] | 70.8 | 65.8 |
| OSS (ours) | 70.4* | **66.1** |

## 4.3 Transfer to other vision tasks

We now test transferability of ImageNet pretrained representation to different vision tasks. In order to better pin-point the gain contributed by each pretraining method, we chose small-scale datasets such as PASCAL VOC [21] and small subsets of COCO [33]. It is because a large amount of data with long finetuning schedule reduces the importance of the initial model. For COCO data, we randomly sample five non-overlapping 1K and 10K train splits as in [28], and the results are averaged over the splits (see B.5 for a sensitivity analysis). We then transfer 800-epoch trained ResNet-18 backbones to object detection on VOC07 with Faster-RCNN [46], semantic segmentation on VOC12 with FCN [34], and object detection and instance segmentation on COCO splits with Mask R-CNN [27] (see A.5, A.6, and A.7 for details). Table 3 shows that our student produces better results than top-performing self-supervised and offline distilled models across all network architectures, datasets, and tasks. Notably, OSS is the only method that outperforms the supervised baselines on COCO-1K and VOC12. More downstream performance on Cityscapes [17] is illustrated in B.4.

## 5 Ablation Studies

### 5.1 Various teacher-student combinations

**Smaller students.** We validate our framework with various teacher-student combinations. Specifically, we consider a ResNet-50 teacher and three small students: MobileNet-v2 [48], RegNetY-600MF [45], and EfficientNet-B0 [51]. We pretrain the three pairs for 200 epochs on ImageNet [19] trainset and conduct linear probe on the validation set (see A.1 and A.3 for details). Table 4 shows that our

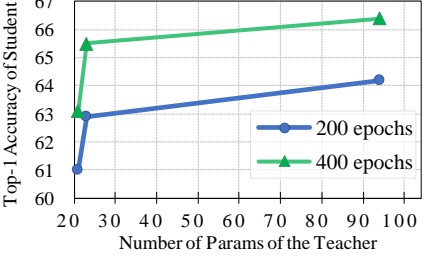

| Teacher | Pretrain Epochs | Teacher Params | Top-1 Teacher | Top-1 Student |
|---|---|---|---|---|
| ResNet-34 | | 21.5M | 62.1 | 61.0 |
| ResNet-50 | 200 | 23.9M | 70.8 | 62.9 |
| ResNet-50w2 | | 94.0M | 75.1 | **64.2** |
| ResNet-34 | | 21.5M | 64.6 | 63.1 |
| ResNet-50 | 400 | 23.9M | 72.0 | 65.5 |
| ResNet-50w2 | | 94.0M | 75.5 | **66.4** |

Figure 3: ImageNet Top-1 accuracy (%) of linear evaluation of ResNet-18 distilled from various teachers. The third column (Params) is the number of parameters in the teacher network, and the unit M is $10^6$. Our student benefits more from larger teacher and longer training schedule.

Table 6: Effect of proposed dropout and distillation schemes on the classification accuracy (%) of ResNet-18 features on ImageNet. Dropout strategy combined with feature classifier improves label efficiency, and online distillation works better than offline version.

| Method Name | Scheme Name | Dropout Applied | Feature Classifier | Top-1 Accuracy Frozen | 1% | 10% | Top-5 Accuracy Frozen | 1% | 10% |
|---|---|---|---|---|---|---|---|---|---|
| Ours | off.KD-Drop | √ | | 66.0 | 33.9 | 61.5 | 86.9 | 59.4 | 84.2 |
| | KD | | | 68.2 | 42.5 | 63.2 | 88.2 | 69.0 | 85.4 |
| | KD-Drop | √ | | **68.4** | 47.2 | 63.3 | **88.5** | 73.3 | 85.6 |
| | KD-Drop-Adv | √ | √ | 68.3 | **47.8** | **63.6** | 88.4 | **73.8** | **85.8** |
| SwAV [6] | Original | | | **64.9** | 39.7 | **60.4** | **86.3** | 66.3 | 83.6 |
| | Drop | √ | | 64.2 | **43.7** | 60.3 | 85.9 | **70.1** | **83.7** |

method outperforms SwAV [6] for all student architectures while the gain contributed by OSS is not equal on different networks. One possible reason is that we use SwAV's hyperparameters, so this may place SwAV at an *advantage*. We also compare our approach with other offline distillation methods using MobileNet-v2 provided in [31]. In Table 5, our MobileNet-v2 features perform better than those compressed from MoCo-v2 [11]'s ResNet-50 teacher even if our teacher's accuracy is lower during training. More results for EfficientNet-B0 encoder are found in B.1, and semi-supervised learning performance is summarized in B.6.

**Larger teachers.** We perform an analysis with ResNet-18 students distilled from three different resnet teachers: ResNet-34, ResNet-50, and ResNet-50w2. ResNet-50w2 is a 50-layer resnet with $2\times$ wider channels. Here, we only consider resource-friendly pairs that are trainable on a single 8-GPU machine, so we do not explore beyond ResNet-50w2. We pretrain them using our framework on ImageNet trainset for 800 epochs and carry out inear probe on the validation set at 200 and 400 epochs. Figure 3 shows that our method benefits more from bigger teachers and longer train iterations. As in supervised knowledge distillation, student's performance is not necessarily proportional to the gap (in size) between teacher and student [36].

## 5.2 Framework design

**Questions and experiment settings.** A natural question that arises is how effective is "offline version of the proposed method", where a student is learning on SwAV's framework with dropout while the target code comes from SwAV's pretrained (frozen) teacher? Also, one might be curious about the performance gain from each proposed technique: network driven multi-view generation and feature classifier. To answer these, we train ResNet-50 (teacher) and ResNet-18 (student) pair under the 4 settings in Table 6 and carry out linear probe and semi-supervised evaluation (see A.3 and A.4). The distillation epoch is 130 for the offline scheme (off.KD-Drop) from SwAV's 800-epoch teacher following [31] and 800 for the other online versions. Implementation details of off.KD-Drop are available in A.8. Throughout this experiment, the dropout rates are 0.1 and 0.05 at the two liner layers in the projection head (see Figure 2).

Table 7: Classification accuracy (%) of ResNet-50 finetuned on small subsets of ImageNet. Results of OSS come from training with ResNet-18 student. The proposed framework produces data efficient teacher as well.

| Method | Top-1 Accuracy 1% | 10% | Top-5 Accuracy 1% | 10% |
|---|---|---|---|---|
| Supervised | 25.4 | 56.4 | 48.4 | 80.4 |
| SimCLR | 48.3 | 65.6 | 75.5 | 87.8 |
| BYOL | 53.2 | 68.8 | 78.4 | 89.0 |
| SwAV | 53.9 | 70.2 | 78.5 | 89.9 |
| OSS (ours) | **57.5** | **71.1** | **81.4** | **90.5** |

Table 8: Effects of dropout rates on low-shot learning classification accuracy on ImageNet for ResNet-18 student. Strong dropout improves data efficiency on 1% setup while performance is less sensitive at 10% setup.

| Dropout rate Linear1 | Linear2 | Top-1 Accuracy 1% | 10% |
|---|---|---|---|
| 0.00 | 0.05 | 37.5 | 60.0 |
| 0.10 | 0.05 | 43.5 | 59.6 |
| 0.10 | 0.10 | 43.7 | 59.5 |
| 0.20 | 0.05 | 44.0 | 59.1 |
| 0.20 | 0.10 | 44.0 | 59.4 |

| Original Image | Supervised Student | offline KD-Drop | | KD | | KD-Drop | | KD-Drop-Adv | |
|---|---|---|---|---|---|---|---|---|---|
| | | Teacher | Student | Teacher | Student | Teacher | Student | Teacher | Student |

Figure 4: Grad-CAM [49] visualization of feature maps of different dropout and distillation schemes using some images in ImageNet validation set. Proposed method (KD-Drop-Adv) shows strongest activation on the target class region (Meatloaf, Marmoset and Bobsled) among all schemes.

**Student's performance analysis.** From Table 6, we observe that online schemes achieve higher accuracy than the offline version (off.KD-Drop). Our intuition is that teachers better figure out and distill appropriate information in the course of joint training by solving the same task online. Yet, our offline method even outperforms all offline techniques in Table 1 in linear evaluation, and this shows superiority of our overall framework. On the other hand, strong performance of our method does not come free from (online) knowledge distillation. We argue that the proposed network driven strategy brings beneficial gain in mining of rich knowledge contained in the network. Indeed, applying random dropout significantly increases data efficiency as demonstrated in 1%-label evaluation. Also, there is evidence that our feature classifier generally yields additional small improvement in low-shot learning performance. From this empirical analysis, we find that the dropout strategy, combined with the feature classifier, is more effective in lower data regime.

**Teacher's performance analysis.** It is natural to ask whether the proposed network driven view generation with feature classifier produces a more data-efficient teacher as the same setup is applied to the teacher network as well. We answer this question by performing the label efficiency evaluation of teacher on ImageNet. We compare the results with those of ResNet-50 models obtained from SSL methods in Table 7. Our teacher outperforms the state-of-the-art self-supervised networks on both 1% and 10% regimes. We note that the training objective of our ResNet-50 teacher is exactly the same as SwAV's ResNet-50 model except for dropout and feature classifier. This verifies that the performance gain over SwAV (+3.6% on 1% set and +0.9% on 10% set in top-1 accuracy) is sorely contributed by our proposed strategy.

**Dropout rates.** In order to provide more intuition on our network driven generation paradigm, we run experiments over multiple choices of dropout rates in the proposed OSS framework (KD-Drop-Adv). We do not explore aggressive dropout rates because self-supervised training signal is already noisy. Table 8 summarizes low-shot classification results on ImageNet. We find that strong dropout improves label efficiency at 1% regime. Yet, accuracy is less sensitive to the choice of dropout probabilities when relatively larger amount of data is available.

**Qualitative analysis of feature distributions.** We visualize feature maps of some images in ImageNet validation set using Grad-CAM [49] in Figure 4. Grad-CAM identifies regions where the network focuses on. We show the results of teacher and student networks trained by our framework with different schemes in Table 6. As in Figure 4, the proposed method (KD-Drop-Adv) shows stronger activation on the target class region than other schemes. While the feature classifier seems to have a little effect on the quantitative performance, the feature map illustrates a clear improvement (over KD-Drop) in the coverage of useful object region. Different from other setups, the student from KD-Drop-Adv reveals high activation on both chunks of meat and wider part of the bobsled and monkey. From this, we see training with the feature classifier helps learning the true data distribution better. More Grad-CAM examples are available in B.8.

### 5.3 Dropout applied to SwAV

We also report the effect of applying our dropout strategy on the performance of SwAV [6] features. We train a dropout added version of SwAV for 800 epochs using a ResNet-18 model. We apply the same dropout scheme to the projection head and compare it with the original model. As shown in Table 6, dropout improves its label efficiency as well.

## 6 Conclusion

We introduce a unified unsupervised visual representation learning framework for small models that combines deep self-supervised learning with knowledge distillation. Our student outperforms the state-of-the-art self-supervised shallow models as well as the networks obtained from top-performing offline compression techniques. We hope our work will attract community's attention to the development of unsupervised pretraining methods for light-weight networks.

### Disclosure of Funding

We received no third party funding for this work.

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
