# A Implementation Details

## A.1 Unsupervised training of our framework

We provide a pseudo code for training our framework in `pytorch` [42] style. The proposed framework is developed upon [6].

```
# teacher model: f (backbone) + h_T (projection head) + L2 normalize
# student model: g (backbone) + h_S (projection head) + L2 normalize
# prototype: C   #E×K
# feature classifier: D (connected with a gradient reversal layer)
# τ and ε are given

for x in loader:
    x_1, x_2 ~ RandomAugmentation(x) and
    x̂_1,...,x̂_V ~ RandomAugmentationSmallCrop(x)

    # embeddings
    z = normalize(h_T(f(cat(x_1,x_2,x̂_1,...,x̂_V))))  #B(2+V)×E
    w = normalize(h_S(g(cat(x_1,x_2,x̂_1,...,x̂_V))))  #B(2+V)×E
    z̃ = normalize(Dropout(h_T(f(cat(x_1,x_2,x̂_1,...,x̂_V)))))  #B(2+V)×E
    w̃ = normalize(Dropout(h_S(g(cat(x_1,x_2,x̂_1,...,x̂_V)))))  #B(2+V)×E

    # dot product of embedding and prototype #B(2+V)×K
    scores_z, scores_z̃ = mm(z, C), mm(z̃, C)
    scores_w, scores_w̃ = mm(w, C), mm(w̃, C)

    loss1, loss2 = 0, 0
    for i in [1,2]:
        with torch.no_grad():
            out = scores_z[B*(i-1):B*i]       # pick up scores
            out = cat(mm(queue[i-1], C), out) # append features

            #fill the queue
            queue[i-1, B:] = queue[i-1, :-B].clone()
            queue[i-1, :B] = z[B*(i-1):B*i]

            #compute codes (use SwAV's sinkhorn func)
            q_i = sinkhorn(exp(out/ε))[-B:]

        # cluster prediction
        subloss1, subloss2 = 0, 0
        for v≠i and v ∈ {1,...,2(V+2)}:
            p_z = Softmax(cat(scores_z, scores_z̃)[B*(v-1):B*v]/τ)
            p_w = Softmax(cat(scores_w, scores_w̃)[B*(v-1):B*v]/τ)
            subloss1 -= mean( sum(q_i * log(p_z)) )
            subloss2 -= mean( sum(q_i * log(p_w)) )
        loss1 += subloss1 / (2*(2+V)-2) / 2
        loss2 += subloss2 / (2*(2+V)-2) / 2

    d_T, d_S = D(cat(z, z̃)), D(cat(w, w̃))
    # Note: feature label is 0 for teacher and 1 for student
    loss = loss1 + loss2 + CrossEntropy(d_T, 0), CrossEntropy(d_S, 1)
    loss.backward()
    update(f.params, g.params, D.params, h_T.params, h_S.params, C)

    # normalize prototypes
    with torch.no_grad():
        C = normalize(C, dim=0, p=2)
```

Full implementation details are described as follows.

**Training hyperparameters.** We implement in our framework with the most of training hyperparameters directly taken from small batch training of [6]. We train our model with stochastic gradient decent using typical batches of 256. We distribute the batches over 4 V100 32GB GPUs, resulting

in each GPU processing 64 images. Synchronized batch norm layers implemented in `pytorch` are used across GPUs. We apply a weight decay of $10^{-6}$, LARS [59] optimizer and a learning rate of 0.6 decayed to a final value of 0.0006 with cosine learning rate schedule.

**Multi-view generation.** We use the data augmentation strategy and default multi-crop setting of [6]: $2 \times 224 + 6 \times 96$, i.e., two $224 \times 224$ full resolution views and six additional $96 \times 96$ small crops. Dropout is applied to projection heads of both teacher and student networks for multi-view generation. Each projection head consists of a linear layer of size 2048 (say, Linear1), followed by batch-norm and ReLU layers and another linear layer of size 128 (say, Linear2). Therefore, each embedding vector is 128-dimensional. We search over dropout rate combinations of $(0.0, 0.05), (0.1, 0.05), (0.1, 0.1), (0.2, 0.05)$, and $(0.2, 0.1)$ at $(\mathrm{Linear1}, \mathrm{Linear2})$. We do not exploit more aggressive dropout rates as self-supervised training signals are already noisy and too much information may be lost with higher rates. Finally, we use the best dropout rates of $(0.1, 0.05)$ at $(\mathrm{Linear1}, \mathrm{Linear2})$.

**Feature classifier.** The feature classifier consists of a hidden linear layer of size 100, followed by batch normalization and ReLU layers, and an output linear layer of size 2. For both teacher and student networks, the feature classifier is connected to the last layer of the projection head via a gradient reversal layer [23] that scales the gradient by a certain negative constant $\alpha$. The constant $\alpha$ is computed using the current training step given by $\alpha = -\frac{2}{1+\exp(-10n/N)} + 1$, where $n$ is the current training step and $N$ is the maximum number of training steps.

**Online clustering.** A queue of stored embeddings of two full resolution images in 15 previous batches is used from the 15th training epochs. The queue has a dimension of $2 \times 128 \times 3840$ as we work with a batch size of 256. As suggested in [6], we augment the stored features when solving the code prediction problem. However, we only use the codes of the current batch for loss calculation. We apply 3 iterations for computing codes online with Sinkhorn algorithm [18] and its regularization parameter $\epsilon$ is set to 0.05. We train our framework with 3000 prototypes (i.e., $K = 3000$).

**Other settings.** The temperature parameter $\tau$ in equation (3) is set to 0.1. We train models for 100, 200, 400 and 800 epochs on ImageNet [19] training set.

## A.2 Existing self-supervised methods

For self-supervised baselines, we re-implement 4 recent top-performing algorithms in `pytorch` [42]: SimCLR [8], MoCo-v2 [11], BYOL [25], and SwAV [6]. We train them for 100, 200, 400 and 800 epochs with the same batch size of 256 on ImageNet [19] training set. For MoCo-v2 and SwAV, we use their official implementation with authors' hyperparameters. We reproduce BYOL and SimCLR using unofficial implementation of `OpenSelfSup` (Apache 2.0 license) [16] and use the same settings described in the corresponding paper. We train two versions of BYOL: one with gradient accumulation (GA) upto 4096 samples (see Appendix G2 of [25] for details) and the other without GA. For a fair comparison with other methods, we use the latter for label efficiency and transfer performance evaluation as all other methods do not accumulate gradients.

## A.3 Linear probe evaluation

We verify our method by linear evaluation on *frozen* features, following the protocol described in [6]. We freeze the unsupervised pretrained final feature representation, and train a supervised linear classifier that consists of a global average pooling layer and a fully-connected layer followed by softmax. We apply standard spatial data augmentation with default parameter settings of `RandomResizedCrop(224)` in `torchvision` library and random horizontal flips during training. Therefore, the input image has a size of $224 \times 224$ for training. At inference time, we first resize the image to $256 \times 256$ along the shorter side and use its center crop of a size $224 \times 224$. The linear classifier is trained for 100 epochs using ImageNet [19] trainset with a batch size of 256, a learning rate of 0.3, cosine learning rate decay, and a weight decay of $10^{-6}$. When training linear classifiers of competing self-supervised models, we sweep over candidate hyperparameters provided by authors and pick the best one.

## A.4 Finetune with 1% or 10% labels

We reproduce the semi-supervised protocol of [6], and use the 1% and 10% splits of ImageNet [19] trainset specified in the officially release of [8]. We train models for 20 epochs with a batch size of 256, and we use different learning rates for feature weights and the classifier weights. For 1% finetuning, we use a learning rate of 0.02 for the feature weights and 5 for the classifier weights. For 10% finetuning, we set a learning rate of 0.01 for the feature weights and 0.2 for the classifier weights. A learning rate decay factor is 0.2 at 12 and 16 epochs, and weight decay is not applied during finetuning. Standard spatial data augmentation strategies (`RandomResizedCrop(224)` in `torchvision` library and random horizontal flips) are applied for training. For evaluation, each image is resized to $256 \times 256$ along the shorter side, and its $224 \times 224$-sized center crop is used. The supervised baselines of 1% and 10% regimes are implemented following the settings in [61]. For each top-performing self-supervised method, we use its optimal settings (for ResNet-50) provided by authors.

## A.5 PASCAL VOC object detection

We finetune ResNet-18 features using Faster R-CNN [46] object detector on VOC2007 dataset [21]. The training is conducted on the official VOC2007 `trainval` set, and the test is performed on the VOC2007 `test` set. As in [29], we use ResNet-18 up to $\text{conv}_4$ stage backbone and the box prediction head implemented in `Detectron2` (Apache 2.0 license) [54] designed for ResNet-50 with appropriately adjusted input channel dimensions. We follow the same finetuning protocol in [29]. That is, we keep the same base learning rate of 0.02, a batch size of 16, total training epoch of 24K, and learning rate decay schedule of 18K and 22K with a decay factor of 0.1 for training. The results are averaged over 5 random seed runs.

## A.6 FCN semantic segmentation

We evaluate our method on VOC2012 [21] and Cityscapes [17] semantic segmentation tasks using FCN [34] in Section 4.3 and B.4. For both experiments, we finetune on the official `trainset` and report results on the `valset`. We use the default implementation settings in `MMSegmentation` (Apache 2.0 license) [15]. The backbone consists of ResNet-18 convolutional layers. This is followed by a decode head that consists of two `Conv-BN-ReLU` blocks with 128 channels. Here, `Conv-BN-ReLU` block consists of a $3 \times 3$ convolutional layer, followed by batch normalization and ReLU activation layers. The output of the decode head is concatenated with the backbone features, and it is followed by one `Conv-BN-ReLU` block with 128 channels and an $1 \times 1$ convolution for per-pixel classification. An auxiliary head is used, and it consists of one `Conv-BN-ReLU` block with 64 channels connected to the $\text{conv}_4$ stage backbone. This is followed by an $1 \times 1$ convolution for per-pixel classification, and the standard per-pixel softmax cross-entropy loss is used for both heads. For both datasets, we apply spatial data augmentations such as random scaling, cropping, and horizontal flipping. The crop size is $512 \times 512$ for VOC2012, and $512 \times 1024$ for Cityscapes. We use a batch size of 16, a weight decay of 0.0005 and a learning rate of $10^{-2}$ with linearly decayed to $10^{-4}$ for VOC2012 and $5 \times 10^{-3}$ for Cityscapes. As suggested in previous work of [29], VOC2012 results are averaged over 5 random seed runs.

## A.7 Mask R-CNN instance segmentation and object detection

**Architecture.** We use Mask R-CNN [27] implemented in `MMDetection` (Apache 2.0 license) [7] and follow the default resnet setups. We use ResNet-18 up to $\text{conv}_4$ as a backbone and apply the same structures with the ResNet-50-FPN for necks and heads except for the input channels of the necks. In particular, we change the input channels to $[64, 128, 256, 512]$ for ResNet-18.

**COCO experiments.** We follow the low-shot learning protocol of [28]. We perform finetuning on randomly sampled five non-overlapping 1K and 10K train splits of COCO [33] dataset, and the average performance on the validation set is reported. The total training epochs are 296 for 10K split and 740 for 1K split, and we use stochastic gradient decent (SGD) optimizer with a learning rate of 0.02, a weight decay of 0.0001 and a momentum of 0.9. The learning rate is reduced at 200 and 264 epochs for 10K and 500 and 660 epochs for 1K with a decay factor of 0.1. We use 4 Tesla M40 GPUs

for training with a total batch size of 16 (4 images per GPU). The performance measure is $AP_{50}$ for detection and instance segmentation tasks.

**Cityscapes experiments.** We test transferability of pretrained features on Cityscapes [17] `gtFine` dataset. We finetune the model using SGD optimizer with a learning rate of 0.02, a weight decay of 0.0001 and a momentum of 0.9. The learning rate is reduced at 8 and 11 epochs with a decay factor of 0.1. We use 4 Tesla M40 GPUs for training with a total batch size of 8 (2 images per GPU).

### A.8   Offline version of the proposed method

We provide implementation details of "offline version of the proposed method" in ablation studies (Section 5.2). Our student ResNet-18 is learning on SwAV [6] framework while the target code comes from SwAV's (800-epoch ImageNet [19] pretrained) *frozen* ResNet-50 teacher. We freeze teacher's backbone and projection head, and prototype vectors while weights in student's backbone and projection head are updated according to gradients from student's loss $L_S$ (see Section 3.1 and Section 3.2 for $L_S$). Random dropout is applied at linear layers in student's projection head with dropout rates of $(0.1, 0.05)$ at $(\text{Linear1}, \text{Linear2})$. The distillation epoch is 130 following [31], and we use 256 batches on 4 V100 32GB GPUs. The target code is computed using Sinkhorn-Knopp algorithm [18] under an equi-partition constraint for each augmented mini-batch with stored embeddings as described in A.1. Here, the embedding vectors are from teacher network. We use the data augmentation and multi-crop strategy and training hyperparameters of OSS implementation (A.1): a weight decay of $10^{-6}$, LARS [59] optimizer and a learning rate of 0.6 decayed to a final value of 0.0006 with cosine learning rate schedule.

## B   Additional Results

### B.1   Offline knowledge distillation methods

We compare our framework with various offline distillation methods that leverage self-supervised teachers in [31] and [22]. We first summarize the techniques therein. CC [40] method is similar to [57] that improves self-supervised learning by distilling quantized teacher's representation. CRD [52] directly compares the feature embeddings of teacher and student and maximizes the mutual information. Reg-BN is a batch-norm augmented version of Reg [47] which regresses the embedding features for distillation. $l$-2 Distance minimizes squared $l$-2 distance. $K$-Means and Online Clustering indicate $K$-means clustering and its online version. "Binary Contr." is a combination of CRD [52]

Table 9: Top-1 accuracy (%) of linear probe evaluation on ImageNet for ResNet-18, MobileNet-v2 and EfficientNet-B0 distilled from variants of self-supervised ResNet-50 (SimCLR[8], MoCo-v2[11], and SwAV[6]). The results are collected from [31] and [22]. T and S denote teacher and student, and * indicates distillation from ResNet-101 teacher. Our method outperforms all competing offline compression techniques.

| Distillation Method | T | T Top-1 | S Top-1 | Distillation Method | T | T Top-1 | S Top-1 |
|---|---|---|---|---|---|---|---|
| Student: ResNet-18 [26] | | | | Student: MobileNet-v2 [48] | | | |
| CC [40] | [6] | 75.6 | 60.8 | CC [40] | [11] | 70.8 | 59.2 |
| CRD [52] | [6] | 75.6 | 58.2 | CRD [52] | [11] | 70.8 | 54.1 |
| Reg-BN [31] | [6] | 75.6 | 60.6 | Reg [60] | [11] | 70.8 | 48.0 |
| $l$-2 Distance [22] | [11] | 67.4 | 55.3 | Reg-BN [31] | [11] | 70.8 | 62.3 |
| K-Means [22] | [11] | 67.4 | 51.0 | CompRess-2q[31] | [11] | 70.8 | 63.0 |
| Online Clustering [22] | [11] | 67.4 | 56.4 | CompRess-1q[31] | [11] | 70.8 | 65.8 |
| Binary Contr. [22] | [11] | 67.4 | 57.4 | Ours | OSS | 70.4 | **66.1** |
| SEED [22] | [8] | 65.6 | 57.5 | Student: EfficientNet-B0 [51] | | | |
| SEED [22] | [11] | 71.1 | 60.5 | | | | |
| CompRess-2q[31] | [6] | 75.6 | 62.4 | SEED [22] | [11] | 67.4 | 61.3 |
| CompRess-1q[31] | [6] | 75.6 | 65.6 | SEED [22] | [11] | 70.3* | 63.0 |
| Ours | OSS | 73.0 | **68.3** | Ours | OSS | 70.9 | **64.1** |

Table 10: Top-1 accuracy (%) of linear probe evaluation on ImageNet for ResNet-18 distilled from ResNet-50 using various distillation epochs. P-E/D-E represent the pretraining epochs of teacher and distillation epochs, and * indicates distillation with additional small patches.

| Teacher | P-E | D-E | Student Top-1 | Teacher | P-E | D-E | Student Top-1 |
|---------|-----|-----|---------------|---------|-----|-----|---------------|
| Distillation Method: SEED [22] | | | | Distillation Method: OSS | | | |
| SwAV [6] | 800 | 100 | 61.1 | Ours | 100 | 100 | 60.0 |
|          | 800 | 200 | 61.7 |      | 200 | 200 | 64.1 |
|          | 800 | 400 | 62.0 |      | 400 | 400 | 65.8 |
| SwAV* [6] | 800 | 200 | **62.6** |   | 800 | 800 | **68.3** |

and contrastive loss. Recent top-performing methods are CompRess [31] and SEED [22], and their training mechanism is similar. SEED maintains a memory bank (queue) for storing data samples' encoding outputs from teacher. SEED calculates instance-level similarity scores between input features extracted from the teacher encoder and all samples in the queue. Similarity scores for a student model with all instances in the queue are computed in the same way, and the student is trained to mimic teacher's score distribution. On the other hand, CompRess decouples teacher and student embeddings and maintain a separate queue for each. There are two implementation versions of CompRess depending on the number of queues therein. If teacher's queue is used in calculating similarity scores for both teacher and student models (as in SEED), it is called CompRess-1q. If each score distribution is computed from the corresponding queue, it is called CompRess-2q. The size of queue in CompRess is 128,000, which is almost double than the size of SEED, 65,536. Comparison results of aforementioned approaches on ResNet-18, MobileNet-v2 [48] and EfficientNet-B0 [51] are illustrated in Tables 9. Overall, our student models outperform all the offline distilled networks even from stronger teachers.

We further compare our method with SEED at various distillation epochs. For both SEED and OSS, we consider a ResNet-50-ResNet-18 (teacher-student) pair, and the corresponding linear probe results (top-1 accuracy) are taken from Figure 1 and SEED's Table 3. As shown in Table 10, SEED works slightly better at 100 epochs, but our model significantly outperforms SEED at 200 and 400 epochs. Also, SEED attains the best accuracy, 62.6%, at 200-epoch distillation using additional small patches. However, this still does not match our 200-epoch performance (64.1%). Marginally worse performance of our method at early stage of training (100 epochs) seems to be due to online joint optimization of teacher and student while SEED is distilled from an already mature teacher. In short, our student learns features a bit slowly at the beginning while eventual performance is significantly better.

## B.2 Transfer performance on other classification tasks

We test whether the features trained on ImageNet [19] with the proposed method are also useful in different image domains via feature *reuse*. We evaluate our ResNet-18 representation by training a

Table 11: Top-1 accuracy (%) of linear probe evaluation for ResNet-18 on various datasets. ResNet-18 features are pretrained on ImageNet trainset, and a linear classifier on top of the frozen representation is trained on CIFAR, SUN397, DTD, Places365 and STL10 datasets.

| Method | Top-1 Accuracy | | | | | |
|--------|---------|----------|--------|-----|-----------|-------|
|        | CIFAR10 | CIFAR100 | SUN397 | DTD | Places365 | STL10 |
| Supervised | **85.3** | **67.0** | 51.8 | 60.2 | 43.4 | 94.8 |
| SimCLR [8] | 79.0 | 54.9 | 50.8 | 63.2 | 42.8 | 89.8 |
| MoCo-v2 [11] | 69.8 | 33.6 | 31.3 | 48.1 | 37.5 | 89.4 |
| BYOL [25] | 60.8 | 25.1 | 24.1 | 48.8 | 33.6 | 84.8 |
| SwAV [6] | 81.4 | 61.4 | 59.3 | 69.9 | **46.6** | 94.8 |
| OSS (ours) | 82.9 | 61.6 | **60.2** | **70.0** | 46.5 | **96.1** |

Table 12: Transfer performance (%) of ImageNet pretrained ResNet-18 features on Places365 low-shot learning tasks. Mean $\pm$ std over 5 splits is given.

| Method | Top-1 Accuracy | |
| | 1% | 10% |
| --- | --- | --- |
| Scratch | 0.3 $\pm$0.04 | 37.4 $\pm$0.21 |
| Supervised | 0.4 $\pm$0.11 | 45.4 $\pm$0.07 |
| SimCLR [8] | 30.6 $\pm$0.35 | 46.2 $\pm$0.10 |
| MoCo-v2 [11] | 32.0 $\pm$0.24 | 46.8 $\pm$0.07 |
| BYOL [25] | 29.8 $\pm$0.17 | 44.9 $\pm$0.13 |
| SwAV [6] | 30.4 $\pm$1.57 | **47.1** $\pm$0.05 |
| CompRess-1q [31] | 34.0 $\pm$0.15 | **47.1** $\pm$0.06 |
| OSS (ours) | **35.1** $\pm$0.25 | **47.1** $\pm$0.07 |

Table 13: Transfer performance (%) of ImageNet pretrained ResNet-18 features on Cityscapes scene segmentation and instance segmentation tasks.

| Method | mIoU | $AP_{50}^{inst}$ |
| --- | --- | --- |
| Supervised | 61.2 | 57.0 |
| SimCLR [8] | 60.8 | **58.6** |
| MoCO-v2 [11] | 61.0 | 57.0 |
| BYOL [25] | 60.4 | 55.5 |
| SwAV [6] | 58.0 | 58.2 |
| CompRess-1q [31] | 54.4 | 57.8 |
| OSS (ours) | **62.4** | 58.3 |

linear classifier on top of the frozen backbone network using 5 datasets, namely CIFAR [32], SUN397 [56], DTD [13], Places365 [63], and STL10 [14], following the procedure described in A.3. We work with the official splits of training and validation except for DTD. For DTD dataset, we only use the first split out of the total 10 splits as in previous studies [8, 25]. We use 800 epoch pretrained models for all unsupervised methods, and results are summarized in Table 11 using the standard metric, 1-crop top-1 accuracy. We note that our ResNet-18 features even surpass the supervised frozen features (trained on ImageNet) on all benchmarks except for CIFAR. Our approach outperforms state-of-the-art self-supervised methods on most of benchmarks.

## B.3 Data efficiency evaluation on Places365

We evaluate the performance obtained when finetuning (ImageNet [19] pretrained) ResNet-18 models on Places365 [63] for classification with small subsets of trainset using labels. We first randomly sample five non-overlapping 1% and 10% train splits. We then train ResNet-18 models on each split and provide average top-1 accuracy (%) and the standard deviation (std) on Places365 validation set in Table 12. OSS significantly surpasses top-performing self-supervised approaches and the state-of-the-art offline knowledge distillation method CompRess-1q [31] (based on SwAV [6]'s ResNet-50 teacher) on 1% split. With 10% split, our method also works best.

## B.4 Transfer performance on Cityscapes

We transfer ImagNet [19] pretrained ResNet-18 representation on Cityscapes [17] instance segmentation with Mask-RCNN [27] and semantic segmentation with FCN [34]. We use 800-epoch pretrained models, and full experimental details are described in A.7 and A.6. Results are reported in Table 13. With Mask-RCNN, SwAV [6] works slightly better than our model. However, our method is the only unsupervised approach that surpasses the supervised baseline in the scene segmentation task.

## B.5 Sensitivity of train split selection on COCO low-shot transfer performance

In COCO transfer experiments of Section 4.3, models are trained on five non-overlapping data splits of 1K and 10K examples, and the results on the validation set are averaged over the five runs. We further provide the average performance (given in Table 3) along with standard deviations to verify that the performance of our model is not sensitive to the selection of train examples. Table 14 shows that OSS consistently outperforms state-of-the-art unsupervised methods regardless of train split selection.

Table 14: Transfer performance (%) of ResNet-18 finetuned on 1K and 10K train splits of COCO dataset for detection and instance segmentation tasks. Each experiment (1K and 10K) is performed on five random non-overlapping train splits. The average $AP_{50}$ over the five runs $\pm$ standard deviation is reported, and the standard deviation is illustrated as an error bar.

| Method | COCO-1K | |
| --- | --- | --- |
| | $AP_{50}^{bb}$ | $AP_{50}^{inst}$ |
| Supervised | 17.0 $\pm$0.54 | 15.6 $\pm$0.67 |
| SimCLR [8] | 16.0 $\pm$0.57 | 14.8 $\pm$0.52 |
| MoCo-v2 [11] | 11.0 $\pm$0.30 | 9.9 $\pm$0.38 |
| BYOL [25] | 14.3 $\pm$0.63 | 12.9 $\pm$0.56 |
| SwAV [6] | 18.6 $\pm$0.53 | 17.2 $\pm$0.60 |
| CompRess-1q [31] | 14.8 $\pm$0.33 | 13.5 $\pm$0.47 |
| OSS (ours) | **20.0** $\pm$0.48 | **18.4** $\pm$0.50 |

(a) COCO-1K experiment

| Method | COCO-10K | |
| --- | --- | --- |
| | $AP_{50}^{bb}$ | $AP_{50}^{inst}$ |
| Supervised | 33.6 $\pm$0.13 | 31.1 $\pm$0.17 |
| SimCLR [8] | 33.0 $\pm$0.23 | 30.6 $\pm$0.26 |
| MoCo-v2 [11] | 28.8 $\pm$0.18 | 26.9 $\pm$0.15 |
| BYOL [25] | 30.4 $\pm$0.35 | 28.0 $\pm$0.04 |
| SwAV [6] | 36.8 $\pm$0.42 | 34.4 $\pm$0.45 |
| CompRess-1q [31] | 31.7 $\pm$0.38 | 29.4 $\pm$0.46 |
| OSS (ours) | **37.9** $\pm$0.16 | **35.4** $\pm$0.24 |

(b) COCO-10K experiment

## B.6 Data efficiency evaluation on ImageNet for light networks

We pretrain three small networks: MobileNet-v2 [48] RegNetY-600MF [45], and EfficientNet-B0 [51] with a ResNet-50 teacher on OSS framework for 200 epochs and report top-1 linear evaluation accuracy on ImageNet [19] validation set in Table 4 of Section 5.1. We further provide 1% and 10% low-shot learning performance of pretrained student features on ImageNet validation set (see A.4 for implementation details). Table 15 shows that our method outperforms SwAV by a large margin except for 1%-experiment with EfficientNet-B0. We have used SwAV's optimal settings and did not search over other parameters due to the computational limitation. We believe performance of OSS models could be better with proper hyperparameter tuning.

Table 15: Classification accuracy (%) of linear probe and low-shot learning evaluation on ImageNet for various shallow models. The second column (Params) is the number of parameters in each network, and the unit M denotes $10^6$.

| Model | Params | Method | Top-1 Accuracy | | | Top-5 Accuracy | | |
|---|---|---|---|---|---|---|---|---|
| | | | Frozen | 1% | 10% | Frozen | 1% | 10% |
| RegNetY-600MF [45] | 6.1M | SwAV [6] | 67.0 | 23.3 | 64.1 | 87.8 | 46.0 | 86.5 |
| | | OSS (ours) | **67.4** | **49.9** | **64.7** | **88.3** | **75.6** | **86.9** |
| EfficientNet-B0 [51] | 5.3M | SwAV [6] | 59.3 | **39.3** | 59.9 | 83.2 | **65.8** | 83.7 |
| | | OSS (ours) | **64.1** | 34.4 | **62.8** | **86.1** | 60.3 | **85.8** |
| MobileNet-v2 [48] | 3.4M | SwAV [6] | 63.2 | 36.8 | 56.1 | 84.7 | 63.1 | 80.3 |
| | | OSS (ours) | **66.1** | **39.4** | **57.6** | **86.7** | **65.6** | **81.4** |

## B.7 Train times

We provide unsupervised pretraining lengths measured in minutes per epoch on 4 V100 32GB GPUs with 256 batches. In Table 16, we report train times of ResNet-18 using our implementation with ResNet-50 teacher (A.1) and SwAV [6]'s ResNet-18 (A.2) on ImageNet [19] training set along with top-1 linear evaluation accuracy on the validation set at 200, 400 and 800 pretraining epochs. Here, accuracy is taken from Figure 1. While per epoch train time of our method is longer than SwAV's due to joint training with ResNet-50, our model's 200 epoch performance (64.1%) is almost on par with SwAV's 800 epoch performance (64.9%). Therefore, training ResNet-18 using our framework with ResNet-50 teacher requires much less computation time to achieve comparable results with SwAV's. Moreover, if we train our model longer, our student ResNet-18 outperforms SwAV's final performance by +0.6 at 400 epochs and +3.4 at 800 epochs.

Table 16: Pretrain times of SwAV's ResNet-18 and OSS's ResNet-18 (trained with ResNet-50 teacher) and top-1 linear probe accuracy of frozen features on ImageNet.

| Method | Time (min/epoch) | Pretrain Epochs | | |
|---|---|---|---|---|
| | | 200 | 400 | 800 |
| SwAV | 25.0 | 61.2 | 63.7 | 64.9 |
| OSS (ours) | 57.5 | **64.1** | **65.5** | **68.3** |

## B.8 Qualitative analysis of feature maps

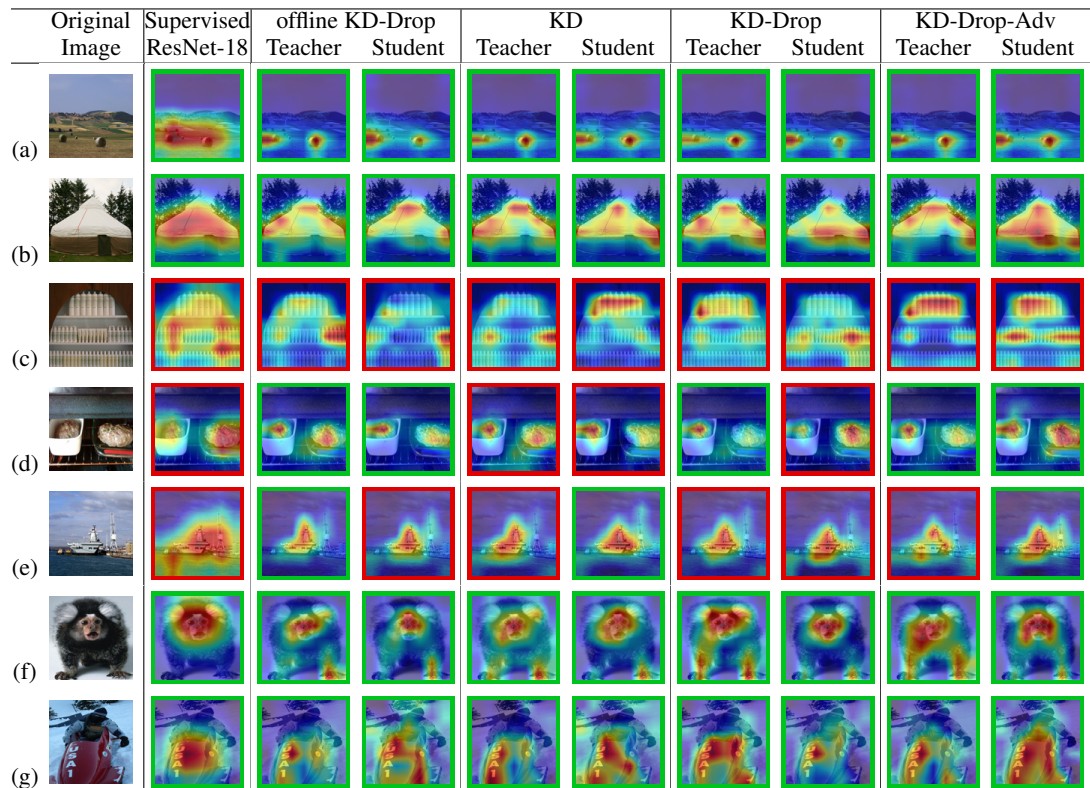

**List of labels and image names**

|     | Label | Label Index | Name | Image Name |
|-----|-------|-------------|------|------------|
| (a) | n07802026 | 816 | hay | ILSVRC2012_val_00043223.JPEG |
| (b) | n04613696 | 714 | yurt | ILSVRC2012_val_00034462.JPEG |
| (c) | n04357314 | 810 | sunscreen, sunblock, sun blocker | ILSVRC2012_val_00006886.JPEG |
| (d) | n07871810 | 806 | meat loaf, meatloaf | ILSVRC2012_val_00035848.JPEG |
| (e) | n02687172 | 246 | aircraft carrier, carrier, flattop | ILSVRC2012_val_00014273.JPEG |
| (f) | n02490219 | 175 | marmoset | ILSVRC2012_val_00015041.JPEG |
| (g) | n02860847 | 252 | bobsled, bobsleigh, bob | ILSVRC2012_val_00033253.JPEG |

Figure 5: Grad-CAM [49] visualization of feature maps corresponding to the ground truth labels for different dropout and distillation schemes. The images are selected from ImageNet validation set. Here, teacher is ResNet-50, and student is ResNet-18. The proposed method (KD-Drop-Adv) shows strongest activation on the target class region (Hay, Yurt, Sunscreen, Meatloaf, Aircraft carrier, Marmoset and Bobsled) among all schemes. Green box boundary indicates the network correctly predicts the class, and red box boundary means the prediction is wrong. Image names and their correct classes are listed in the table.