# OpenReview forum: "Unsupervised Representation Transfer for Small Networks: I Believe I Can Distill On-the-Fly"
_NeurIPS.cc/2021/Conference — NeurIPS 2021 Poster_

### Official Review · Reviewer_Ymks · 2021-07-11

**Rating:** 7
**Confidence:** 4

**Summary:**

This paper presents a self-supervised distillation method that aims to learn better-performing representations for small models.  In particular, the teacher model is trained to produce consistent cluster assignments between different small views of the image while the student model is encouraged to mimic the prediction of the teacher. Comprehensive experiments on ImageNet and other downstream tasks validate the effectiveness of the self-supervised distillation method.

**Limitations And Societal Impact:**

 the authors adequately addressed the limitations and potential negative societal impact of their work.

**Main Review:**

The paper presents a distillation-based self-supervised representation method, which is largely based on the previous efforts of SEED [1] and CompRess [2]. In essence, the author proposes to train both the Teacher and Student architecture, with the Teacher conducting the clustering and providing the soft labels for the Student. The author also proposes to leverage the multi-view strategies similar to [1] and [3]. The whole paper is well-organized, with comprehensive experiments and ablations, and clear paper writing. The exhibited performances are competitive and show great improvement over previous self-supervised distillation baselines and self-supervised methods.

The concerns from the reviewer mainly come from the following aspects of incremental technical novelty comparing to [1]; and the doubts that whether the over-performing numbers come at the great training cost of additional optimizing the bulky Teacher architecture; In addition, it is much preferred to show comparisons with SEED and CompRess instead of SWAV/SimCLR.

Specific comments are listed as follows:

1. About the novelty of contribution: It seems the proposed methods are mostly inspired and originated from [1] and [2]. In particular, SEED also adopted the `"instance dictionary", which I think has the same effect as the prototype clustering. Also, it seems [1]'s clustering is also based on the representations from the Teacher and also the multi-view strategy. Comparing to [1], what are the main differences that distinguish OSS? In other words, what are the reasons that further improve the performances? Is it the Teacher's continuing optimization jointly with the Student?

2. Concerns about Figure 1. The figure demonstrates the advantages of using the self-supervised distillation methods comparing with previous state-of-the-art SSL methods. However, as a consensus, distillation usually leads to efficient convergencies and better performances under the same training epochs. Is it a fair comparison with SSL methods instead of SEED and CompRess?

I strongly suggest the author instead show the numbers w.r.t. to the number of epochs of SEED and CompRESS instead of SWAV/SimCLR.

3. Additional training cost. OSS shows great performance improvement over SSL and [1], [2], however the requirement to keep the Teacher model trainable during the distillation&training. I understand that the longer training epochs exist in most previous SSL baselines, while this also unavoidably narrows the OSS' scope of applications. I.e., given a generic SSL learned Teacher arch, isn't OSS sacrificing the cost of training a cumbersome Teacher for better performances?

Suggestion: report also the offline distillation numbers when the Teacher arch is frozen and not updated. This quantifies the improvements gained from the online/offline setting.


4. It is strongly suggested to append specific discussion about the differences of OSS with CompRess instead of SimCLR when no distillation technique is leveraged. The current draft majorly compares with MoCo/v2 and other SSL methods and over-emphasizes its position in SSL, which I think is a bit one-sided.

Suggestion: focus more on the other self-supervised distillation and reduce the statement of comparison over SSL methods.

Overall, I appreciate the authors with their efforts for comprehensive experiments and the showed performance-boosting is inspiring. I'm very prone to further update my ratings with further responses from the authors and I still hold quite positive opinions for this submission after fixing my concerns.


References

[1]. Zhiyuan Fang, Jianfeng Wang, Lijuan Wang, Lei Zhang, Yezhou Yang, and Zicheng Liu. SEED: Self-supervised distillation for visual representation. In International Conference on Learning Representations (ICLR), 2021.

[2]. Soroush Abbasi Koohpayegani, Ajinkya Tejankar, and Hamed Pirsiavash. CompRess: Self380 supervised learning by compressing representations. In Proceedings of Advances in Neural Information Processing Systems (NeurIPS), 2020.

[3]. Jean-Bastien Grill, Florian Strub, Florent Altché, Corentin Tallec, Pierre H. Richemond, Elena Buchatskaya, Carl Doersch, Bernardo Avila Pires, Zhaohan Daniel Guo, Mohammad Gheshlaghi  Azar, Bilal Piot, Koray Kavukcuoglu, Rémi Munos, and Michal Valko. Bootstrap your own latent: A new approach to self-supervised learning. In Proceedings of Advances in Neural Information Processing Systems (NeurIPS), 2020.

**Time Spent Reviewing:**

4

---

> ### Author Response · Authors · 2021-08-10
> **We thank the reviewer for his/her comments and suggestions.**
>
> **Q1**: Comparing to SEED, what are the main differences that distinguish OSS? What are the reasons that further improve the performances? Is it the teacher's continuing optimization jointly with the student?
>
> **A1**: A key difference between our method and SEED is overall training mechanism. As pointed out by the reviewer, both SEED [1] and our framework maintain an instance queue for storing data samples’ embeddings from teacher. SEED calculates instance-level similarity scores between input features extracted from a static (already trained) teacher encoder and "all samples" in the queue. Similarity scores for a student model with all instances in the queue are computed in the same way, and the student is trained to mimic teacher's score distribution using cross entropy loss. On the other hand, our framework is based on coarser online clustering of input features and the embedding vectors in the queue instead of performing instance-level comparison. Our intuition is that fine instance-level discrimination seems to be more difficult for smaller models with fewer parameters. Our teacher model is trained to predict cluster assignment scores of a view from the representation of another view, and simultaneously, a student is encouraged to mimic teacher's cluster prediction.
>
> We believe that our overall training framework is more effective because even "offline version" of our method works better than SEED (off.KD-Drop (ours): 66.0\% from Table 6 vs SEED: 62.6\% from Table 1 in ImageNet linear evaluation). In addition, further performance improvement especially in data efficiency is due to online joint optimization and feature classifier as shown in Table 6.
>
> ----
> **Q4**: It is strongly suggested to append specific discussion about the differences of OSS with CompRess instead of SimCLR where no distillation technique is leveraged.
>
> **A4**: Overall design of CompRess [2] is similar to SEED [1], so the same description regarding differences between SEED and OSS in the answer to Q1 applies for this case. Indeed, both CompRess and SEED are based on offline knowledge distillation from static self-supervised teacher and store feature embeddings in a queue. Also, instance-level similarity scores between input features and all samples in the queue are computed, and the distance between teacher and student score distributions is minimized using the same cross entropy loss during training.
>
> We now elaborate some differences between SEED and CompRess. Recall that SEED stores its teacher's features in a queue. However, CompRess decouples teacher and student embeddings and maintain a separate memory bank (queue) for each. There are two implementation versions of CompRess depending on the number of queues used therein. If teacher's queue is used in calculating similarity scores for both teacher and student models (like in SEED), it is called CompRess-1q. In this case, the training objective is the same as SEED. On the other hand, if each score distribution is computed from the corresponding queue, it is called CompRess-2q. It has been shown that CompRess-1q works better than CompRess-2q. Another difference between SEED and CompRess is the queue size. The size of each queue in CompRess is 128,000, which is almost double than the size of SEED, 65,536. We note that OSS has a relatively small queue size, $3840\times 2=7,680$.
>
> ----
> **Q2**: Concern about Figure 1. Is it a fair comparison with SSL methods instead of SEED and CompRess? I strongly suggest the author instead show the number w.r.t the number of epochs of SEED and CompRess instead of SwAV and SimCLR.
>
> **A2**: Training framework of SEED [1] and CompRess-1q [2] is mostly similar except for minor settings such as queue size and training hyperparameters (see the answer to Q4), and this is also pointed out by an reviewer of SEED (Reviewer2) at  https://openreview.net/forum?id=AHm3dbp7D1D. Due to time constraints, we perform a comparison of our approach with SEED (more recent work) and will update results of CompRess-1q later.
>
> We compare our method with SEED at 100, 200, and 400 pretraining epochs as 800-epoch accuracy of SEED is not available. For both SEED and our approach, we consider ResNet-50-ResNet-18 (teacher-student) pairs, and corresponding linear probe results (Top-1 accuracy) are taken from Figure 1 and SEED's Table 3. Here, SEED models are distilled from (static) SwAV [3]'s 800-epoch pretrained teacher. Below table shows that SEED works slightly better at 100 epochs, but our model significantly outperforms SEED at 200 and 400 epochs. (In the table, P-E/D-E represent the pretraining and distillation epochs.) Moreover, SEED attains the best ResNet-18 Top-1 accuracy, 62.6\% (Table 1), at 200-epoch pretraining with distillation using additional small patches, but this still does not match our 200-epoch performance (64.1\%). Marginally worse performance of our method at early stage of training (100 epochs) seems to be due to online joint optimization of teacher and student while SEED is distilled from an already mature teacher. In short, our student learns features a bit slowly at the beginning, but eventual performance gain over SEED is huge.
>
> In fact, a direct comparison of our model's performance with the reported linear evaluation accuracy by CompRess (65.6\% in Table 1 at 130 epochs) is not fair even if the same epoch models are evaluated. This is because CompRess does not follow the common linear probe protocol used in self-supervised representation learning literature. The common procedure is to add a linear classifier that consists of a single fully-connected layer and a softmax on top of the frozen average pooled backbone feature and report classification accuracy after training the linear layer. On the other hand, CompRess uses two extra normalization layers (one $l_2$-normalization and one batch normalization layer) in the classifier (see Section 4.1 in Compress). For a fair comparison, we are now working on conducting the "common" linear probe evaluation of CompRess model, and we will provide this result later.
>
> | Method      | P-E | D-E | Top-1 |
> |-------------|:---:|:---:|------:|
> | SEED + SWAV | 800 | 100 | 61.1  |
> |             | 800 | 200 | 61.7  |
> |             | 800 | 400 | 62.0  |
> | Ours    | 100 | 100 | 60.0  |
> |             | 200 | 200 | 64.1  |
> |             | 400 | 400 | __65.8__  |
>
> *P-E/D-E represent the pretraining epochs of teacher and distillation epochs.
>
>
> ----
> **Q3**: Additional training cost. Given a generic SSL learned teacher arch, isn't OSS sacrifising the cost of training a cumbersome teacher for better performance? Report also the offline distillation numbers when the teacher arch is frozen and not updated. This quantifies the improvements gained from the online/offline setting.
>
> **A3**: Additional training cost. In order to show performance gain of our online setting over offline, we have reported results of ResNet-18 student that is trained with “offline version of the proposed method" (Off.KD+Drop). Here, the student model is learning from SwAV’s 800-epoch pretrained (frozen) ResNet-50 teacher on our framework with dropout (see Supplemental A.8 for implementation details). The offline distillation epoch was set to 130 (as in CompRess [2]), and Table 6 shows that online model works a lot better than the offline distilled version in linear probe and label efficiency evaluation on ImageNet.
>
> In the below table, we further provide transfer performance ($AP_{50}$ in %) of the above mentioned offline distilled ResNet-18 using some tasks of Table 3 (ResNet-18 finetuned on small-scale datasets for detection and instance segmentation tasks). Note that results of CompRess-1q and OSS are not new and taken from Table 3. We see that the proposed online model outperforms the offline distilled version in all transfer learning tasks. Admittedly, our online framework requires more training time as our teacher is also learning from scratch together. Yet, one may consider using our offline scheme if he or she values more on small training cost than high performance. As shown in the below table, our offline student transfers better than the model from competing offline distillation method of CompRess.
>
> | Method          | VOC07 $bb_{50}$ | COCO-1K $bb_{50}$ | COCO-10K $bb_{50}$ | COCO-1K $inst_{50}$ | COCO-10K $inst_{50}$ |
> |-----------------|------:|--------:|---------:|--------:|---------:|
> | CompRess-1q [2] | 66.4  | 14.8    | 31.7     | 13.5    | 29.4     |
> | off.KD+Drop     | 66.7  | 19.4    | 37.3     | 17.7    | 34.7     |
> | OSS (ours)      | 68.3  | 20.0    | 37.9     | 18.4    | 35.4     |
>
>
> ----
> **References**
>
> [1] Zhiyuan Fang, Jianfeng Wang, Lijuan Wang, Lei Zhang, Yezhou Yang, and Zicheng Liu. SEED: Self-supervised distillation for visual representation. In International Conference on Learning Representations (ICLR), 2021.
>
> [2] Soroush Abbasi Koohpayegani, Ajinkya Tejankar, and Hamed Pirsiavash. CompRess: Self-supervised learning by compressing representations. In Proceedings of Advances in Neural Information Processing Systems (NeurIPS), 2020.
>
> [3] Mathilde Caron, Ishan Misra, Julien Mairal, Priya Goyal, Piotr Bojanowski, and Armand Joulin. Unsupervised learning of visual features by contrasting cluster assignments. In Proceedings of Advances in Neural Information Processing Systems (NeurIPS), 2020.

---

### Official Review · Reviewer_RWpQ · 2021-07-12

**Rating:** 6
**Confidence:** 4

**Summary:**

The authors target an interesting and valuable problem: how to train an unsupervised/self-supervised small model by distillation large models. They proposed the OSS framework, which is equipped with a distillation strategy, a multi-view generation strategy, and an adversarial feature generation strategy. The proposed framework is shown to work better than current SOTA self-supervised frameworks and distillation methods. In some specific regimes, the proposed framework works even better than the supervised models. This work makes a valuable exploration for unsupervised/self-supervised learning with light-weight models.

**Limitations And Societal Impact:**

In the related work, the authors summarized the unsupervised methods into several categories: construction-based methods, prediction-based methods, cluster-based methods and contrastive methods. In this paper, the authors mainly explore the proposed OSS framework on one of the cluster-based methods. More valuable explorations may be applied to other cluster-based methods and also methods on the other categories, at least the contrastive methods may also work.

**Main Review:**

This work tackles an interesting problem of unsupervised learning with light-weight models. The proposed OSS framework is effective and the experimental results could address all the claims of this paper. However, I still have some concerns regarding the model design and experimental settings.

Model Design:
> - For Eq. (5), since $\hat{z}$ are the dropout features and they are derived from $\tilde{z}$ through dropout, and during a long time of training, different versions of $\hat{z}$ could be sampled. And the goal of the multi-view generation is to make the model can equally make use of the multi-view versions of the features. In other words, we want $\hat{z}$ and $\tilde{z}$ can work equally for the model. So, why is it necessary to include a cross-entropy loss term for $\tilde{z}$, or I may want to ask why the sum of $L_{c}(\hat{z})$ and $L_{c}(\tilde{z})$ can work as such a role?
> - For Eq. (7), the adversarial training. How did this loss backward to the main feature models? Shall we use $L_{F} = -\sum \sum (log d_{T}^{(1)} + log d_S^{(0)})$?

Experiments:
> - In Fig. 1 and Table 1, how about the performance of the "vanilla SwAV Teacher" model and the "Supervised Teacher Model" (Baseline)? More results can help to better understanding the results and the improvements.
> - In line 186, the authors mentioned "A plausible reason for this is that large models easily overfit to the training data, so the static teachers provide less extra knowledge beyond hard annotations", so why OSS model is not influenced by this issue?
> - In Table 6, how about the performance of the variant KD-Adv?

Typos:
> - line 2 "greatly have reduced" --> have greatly reduced

**Time Spent Reviewing:**

6

---

> ### Author Response · Authors · 2021-08-10
> **We thank the reviewer for his/her comments and suggestions.**
>
> **Q1**: For Eq. (5), since $\hat{z}$ are the dropout features and they are derived from $\tilde{z}$ through dropout, and during a long time of training, different versions of $\hat{z}$ could be sampled. And the goal of the multi-view generation is to make the model can equally make use of the multi-view versions of the features. In other words, we want $\hat{z}$ and $\tilde{z}$ can work equally for the model. So, why is it necessary to include a cross-entropy loss term for $\tilde{z}$, or I may want to ask why the sum of $L_c(\hat{z})$ and $L_c(\tilde{z})$ can work as such a role?
>
>
> **A1**: We first clarify notations. As described in line 143, $\hat{z}$ denotes small-crop features (not dropout features), and $\tilde{z}$ is dropout-applied features.
>
> Multi-view generation strategy has been used to better learn average representation of given input over the distribution of random augmentation. It is empirically shown that increasing the number of different views during self-supervised representation learning improves the quality of resulting features [1-3]. Along this line, we introduce a new network-driven paradigm with dropout in order to exploit rich feature representation contained in the network itself. In our framework, we use both dropout features $\tilde{z}$ as well as existing spatial augmentation based multi-crop features $\hat{z}$ [3]. Our intuition is that training with both multi-crop and dropout features is more robust. It is because small-crops contain relatively partial information  of inputs (small-crops cover small parts of images) and dropout applied small-crop features might be too hard examples when their non-dropout version of features are not learned together. Note also that for each small-crop $i$ of image $n$, we have $P(\tilde{z}\_{ni} = \hat{z}\_{ni}) = {0.9}^{2048}\times{0.95}^{128}$
> (almost zero probability) with our default dropout rates of $(0.1, 0.05)$ at linear layers (Linear1, Linear2), where Linear1 has size of 2048 and Linear2 has size of 128 (Supplemental A.1). Therefore, we believe current equal weighting for $L_c(\hat{z})$ and $L_c(\tilde{z})$ is reasonable.
>
>
>
> ----
> **Q2**: For Eq. (7), the adversarial training. How did this loss backward to the main feature models?
>
>
> **A2**: For both teacher and student networks, the feature classifier is connected to the last layer of the projection head via a gradient reversal layer [4]. The gradient reversal layer (GRL) has no trainable parameters, and it acts an identity transformation for forward pass. During the backpropagation however, the GRL takes the gradient from the subsequent level and reverse its direction, i.e., multiplies it by some negative constant, before passing it to the preceding layer. Note that "without GRL", optimizer tries to make the teacher and student features dissimilar by minimizing the feature classifier loss. On the other hand, "with GRL", minimizing the loss reduces discriminative power of the feature classifier.
>
> ----
> **Q3**: In Fig.1 and Table 1, how about the performance of the "vanilla SwAV Teacher" model and the "Supervised Teacher Model" (Baseline)? More results can help to better understanding the results and the improvements.
>
>
> **A3**: Top-1 accuracy of SwAV's 800-epoch ResNet-50 (4096 batch training result) is 75.3%, and top-1 accuracy of the supervised ResNet-50 baseline (torchvision model) is 76.1%. We will add this information in the final version.
>
> ----
> **Q4**: In line 186, the authors mentioned "A plausible reason for this is that large models easily overfit to the training data, so the static teachers provide less extra knowledge beyond hard annotations", so why OSS model is not influenced by this issue?
>
>
> **A4**: As the proposed framework is based on online distillation, supervision signals produced by our evolving teacher are softer labels during training. Our intuition is that online teachers better figure out and distill appropriate information in the course of joint training by solving the same task online.
>
> ----
> **Q5**: In Table 6, how about the performance of the variant KD-Adv?
>
> **A5**: Due to computational limitation, we instead carry out linear probe and label efficiency evaluation on ImageNet at (early stopped) 200-epoch pretrained models. We will include the performance of 800-epoch pretrained model with KD-Adv setup in the final version. Below table provides the results (%), and we see similar patterns shown in Table 6. (1) KD vs KD-Drop: Applying random dropout operation significantly improves data efficiency as demonstrated in 1%-label evaluation. (2) KD-Drop vs KD-Drop-Adv: Our feature classifier generally introduces additional small gain in label efficiency. We now describe new findings from KD-Adv combination as follows. (3) KD vs KD-Adv: Our feature classifier is also effective in improving label efficiency. (4) KD-Adv vs KD-Drop-Adv: Dropout added version works better than non-dropout version.
>
> | Scheme           | Dropout | Feature Cls. | Top-1 Frozen | Top-1 1% | Top-1 10% | Top-5 Frozen | Top-5 1% | Top-5 10% |
> |------------------|---------|--------------|--------------|----------|-----------|--------------|----------|-----------|
> | KD               |         |              | 62.8         | 31.7     | 59.3      | 84.5         | 57.4     | 82.7      |
> | KD-Drop          | V       |              | 62.9         | 43.5     | 59.4      | 84.8         | 69.5     | 82.7      |
> | KD-Adv           |         | V            | 62.5         | 42.8     | 59.2      | 84.5         | 69.4     | 82.4      |
> | KD-Drop-Adv      | V       | V            | 62.9         | 43.8     | 59.4      | 85.0         | 69.8     | 83.0      |
>
> ----
> **Q6**: Typo line2.
>
> **A6**: We will correct typos.
>
> ----
> **References**
>
> [1] Mathilde Caron, Ishan Misra, Julien Mairal, Priya Goyal, Piotr Bojanowski, and Armand Joulin. Unsupervised learning of visual features by contrasting cluster assignments. In Proceedings of Advances in Neural Information Processing Systems (NeurIPS), 2020.
>
> [2] Ting Chen, Simon Kornblith, Mohammad Norouzi, and Geoffrey Hinton. A simple framework for contrastive learning of visual representations. In Proceedings of the 37th International Conference on Machine Learning, pages 1597–1607, 2020.
>
> [3] Zhirong Wu, Yuanjun Xiong, Stella X. Yu, and Dahua Lin. Unsupervised feature learning via non-parametric instance discrimination. In IEEE Conference on Computer Vision and Pattern Recognition (CVPR), pages 3742–3733, 2018.
>
> [4] Yaroslav Ganin, Evgeniya Ustinova, Hana Ajakan, Pascal Germain, Hugo Larochelle, François Laviolette, Mario Marchand, and Victor Lempitsky. Domain-adversarial training of neural networks. Journal of Machine Learning Research, 17(59):1–35, 2016.

---

### Official Review · Reviewer_jWLC · 2021-07-19

**Rating:** 6
**Confidence:** 4

**Summary:**

The paper proposes an unsupervised representation learning method. It follows a teacher-student paradigm. The teacher (ResNet-50) is learned by fitting samples to clusters defined by randomly initialized prototypes and fitting different views of the same sample to the same cluster. The views are created with feature dropouts instead of random geometric transformations. The student model (ResNet-18) is distilled from the teacher online, i.e., trained together with the teacher.  Adversarial learning is also applied to align the features between the student and teachers.

**Ethics Review Area:**

["I don’t know"]

**Limitations And Societal Impact:**

The paper is about general unsupervised learning. I did not see particular limitations to address.

**Main Review:**

Pros:

1.	Online distillation is relatively underexplored in the particular setting of unsupervised representation learning.
2.	The proposed method is a reasonable combination of various techniques. Though the separate components are not new, a working combination of them is always valuable.
3.	Experimental results in distilling ResNet50 to ResNet18 demonstrate that the proposed method outperforms previous distillation-based methods significantly and outperforms other unsupervised learning methods trained directly on ResNet18.

Cons:

1.	Online teacher-student update is not new in the teacher-student paradigm.
2.	While the benefits of using a small network are generally justifiable, constraining a method strictly to small networks makes its potential application narrow. The absence of results with larger networks also raises the question of whether the method can work on larger networks at all. The paper could be stronger if it explores settings beyond ResNet50->ResNet18.
3.	The online distillation and adversarial learning ideas are orthogonal to how the other aspect of unsupervised learning. That means such a distillation strategy can be combined with other unsupervised learning methods, such as SimCLR, MOCO, BYOL, SwAV. While outperforming existing offline distillation methods, the proposed method might be a better mate to a different unsupervised learning method.

---

Thanks for the authors' response.


**Time Spent Reviewing:**

2

---

> ### Author Response · Authors · 2021-08-09
> **We thank the reviewer for his/her comments and suggestions.**
>
> **Q1**: Online teacher-student update is not new in the teacher-student paradigm.
>
> **A1**: We agree, but the proposed work is the first study verifying its effectiveness in the context of unsupervised representation learning over multiple network architectures and various datasets and vision tasks.
>
> ----
> **Q2**: The absence of results with larger networks also raises the question whether the method can work on larger networks at all. The paper could be stronger if it explores settings beyond ResNet-50-ResNet-18.
>
> **A2**: We conjecture that heavier students also benefit from our method. However, our main focus is on unsupervised representation learning for light networks, so we only considered resource-friendly combinations that are trainable on a single 8-GPU machine. In fact, results on various teacher-student combinations beyond ResNet-50-ResNet-18 have been reported. To be specific, we considered ResNet-50 teacher and three students: MobileNet-V2, RegNetY-600MF, and EfficientNet-B0 in Tables 4 and 5 of Section 5.1 and Table 14 of Supplemental B.7. We also performed an analysis with ResNet-18 student distilled from three different resnet teachers: ResNet-34, ResNet-50, and ResNet-50w2 in Figure 3. Table 9 in Supplemental B.1 provides an extensive list of comparison for ResNet-18, MobileNet-v2 and EfficientNet-B0 distilled using various methods from varying self-supervised ResNet-50 teachers.
>
> ----
>
> **Q3**: A distillation strategy can be combined with other unsupervised learning methods such as SimCLR, MoCo, BYOL, SwAV. While outperforming existing offline distillation methods, the proposed method might be a better mate to a different unsupervised learning method.
>
> **A3**: We agree that a distillation strategy can be combined with other methods. Yet, we believe our SwAV-like clustering based framework is more reasonable because of the following reasons. First, instance-level discrimination task seems to be more difficult for smaller models with fewer parameters. This is pointed out by SEED [1] using MoCoV2 [2] training with light networks such as EfficientNet-B0, EfficientNet-B1, MobileNet-V3, and ResNet-18. Also, in terms of small-batch training performance (assuming limited computational budget), SwAV [3] works better than the other approaches (see Figure 1 for ResNet-18 and Table 3 in SwAV for ResNet-50). Moreover, BYOL [4] and MoCoV2 require extra components. BYOL relies on gradient accumulation technique (upto 4096 batches) to perform small-batch training (see Appendix G2 in BYOL), and MoCo framework is based on two encoders, i.e., query and momentum encoders.
>
> ----
> **References**
>
> [1] Zhiyuan Fang, Jianfeng Wang, Lijuan Wang, Lei Zhang, Yezhou Yang, and Zicheng Liu. SEED: Self-supervised distillation for visual representation. In International Conference on Learning Representations (ICLR), 2021.
>
> [2] Xinlei Chen, Haoqi Fan, Ross Girshick, and Kaiming He. Improved baselines with momentum contrastive learning, 2020. arXiv: 2003.04297.
>
> [3] Mathilde Caron, Ishan Misra, Julien Mairal, Priya Goyal, Piotr Bojanowski, and Armand Joulin. Unsupervised learning of visual features by contrasting cluster assignments. In Proceedings of Advances in Neural Information Processing Systems (NeurIPS), 2020.
>
> [4] Jean-Bastien Grill, Florian Strub, Florent Altche, Corentin Tallec, Pierre H. Richemond, Elena Buchatskaya, Carl Doersch, Bernardo Avila Pires, Zhaohan Daniel Guo, Mohammad Gheshlaghi Azar, Bilal Piot, Koray Kavukcuoglu, Remi Munos, and Michal Valko. Bootstrap your own latent: A new approach to self-supervised learning. In Proceedings of Advances in Neural Information Processing Systems (NeurIPS), 2020.

---

### Decision · Program_Chairs · 2021-09-27

**Decision:**

Accept (Poster)

**Comment:**

All reviewers were supportive of the ideas in the paper and recommended acceptance - I agree with them. I also believe that studying small networks is an important area in itself due to the need for lightweight models on "edge devices". The reviewers have provided a number of valuable comments which the authors should integrate into their final version.